# Mental Health Disorder Symptoms among Canadian Coast Guard and Conservation and Protection Officers

**DOI:** 10.3390/ijerph192315696

**Published:** 2022-11-25

**Authors:** Katie L. Andrews, Laleh Jamshidi, Jolan Nisbet, Taylor A. Teckchandani, Jill A. B. Price, Rosemary Ricciardelli, Gregory S. Anderson, R. Nicholas Carleton

**Affiliations:** 1Canadian Institute of Public Safety Research and Treatment (CIPSRT), University of Regina, Regina, SK S4S 0A2, Canada; 2Fisheries and Marine Institute, Memorial University of Newfoundland, St. John’s, NL A1C 5R3, Canada; 3Faculty of Science, Thompson Rivers University, Kamloops, BC V2C 0C8, Canada

**Keywords:** mental health, public safety personnel (PSP), post-traumatic stress disorder (PTSD), post-traumatic stress injury (PTSI), occupational stressors

## Abstract

Canadian public safety personnel (PSP) screen positive for one or more mental health disorders, based on self-reported symptoms, at a prevalence much greater (i.e., 44.5%) than the diagnostic prevalence for the general public (10.1%). Potentially psychologically traumatic event (PPTE) exposures and occupational stressors increase the risks of developing symptoms of mental health disorders. The current study was designed to estimate the mental health disorder symptoms among Canadian Coast Guard (CCG) and Conservation and Protection (C&P) Officers. The participants (*n* = 412; 56.1% male, 37.4% female) completed an online survey assessing their current mental health disorder symptoms using screening measures and sociodemographic information. The participants screened positive for one or more current mental health disorders (42.0%; e.g., post-traumatic stress disorder, major depressive disorder, generalized anxiety disorder, social anxiety disorder, panic disorder, alcohol use disorder) more frequently than in the general population diagnostic prevalence (10.1%; *p* < 0.001). The current results provide the first information describing the prevalence of current mental health disorder symptoms and subsequent positive screenings of CCG and C&P Officers. The results evidence a higher prevalence of positive screenings for mental health disorders than in the general population, and differences among the disorder-screening prevalence relative to other Canadian PSP. The current results provide insightful information into the mental health challenges facing CCG and C&P PSP and inform efforts to mitigate and manage PTSI among PSP. Ongoing efforts are needed to protect CCG and C&P Officers’ mental health by mitigating the impacts of risk factors and operational and organizational stressors through interventions and training, thus reducing the prevalence of occupational stress injuries.

## 1. Introduction

Public safety personnel (PSP) include, but are not limited to, border services officers, correctional workers, firefighters (career and volunteer), Indigenous emergency management, operational and intelligence personnel, paramedics, policing (municipal and provincial), public safety communication members, Royal Canadian Mounted Police, and search and rescue personnel [1]. At least two operating agencies include PSP within the Department of Fisheries and Oceans Canada (DFO): specifically, the Canadian Coast Guard (CCG), and Conservation and Protection Officers (C&P).The CCG include PSP with duty-specific responsibilities that involve ensuring Canada’s sovereignty and security, conducting search and rescue operations, and providing marine assistance across Canadian waters [2,3]. C&P PSP have duty-specific responsibilities related to law enforcement and the protection of species at risk, fish habitats, and oceans, carry out a wide range of duties, both on land and at sea, overtly and covertly, and often encounter confrontational members of the public in remote locations, with little to no backup assistance.

During PSP employment, including CCG and C&P occupational activities, regular exposure to potentially psychologically traumatic events (PPTEs) [1], such as exposure to threatened or actual physical assaults, fires, or explosions, is expected [4,5]. Canadian PSP [4,6], and more recently CCG and C&P Officers [5], have reported statistically significantly more PPTE exposures compared to the general population. Such exposures have been reported to be associated with an increased risk of the development of mental health disorders (e.g., post-traumatic stress disorder [PTSD], major depressive disorder [MDD], panic disorder [PD], generalized anxiety disorder [GAD], social anxiety disorder [SAD], and alcohol use disorder [AUD]) [4,5]. In a diverse sample of Canadian PSP (i.e., municipal/provincial police, firefighters, paramedics, Royal Canadian Mounted Police, correctional workers, and dispatchers), 44.5% screened positive for any mental health disorder; 23.2% screened positive for PTSD; and 26.4% screened positive for MDD [7]. From the same study, female municipal/provincial police and firefighters were more likely than males to report mental health disorder symptoms, which is consistent with the evidence that females in the general population are more likely than men to report mental health disorders [8,9,10]. However, the sex differences were not observed across all PSP groups, suggesting diverse systemic variables differentially affect female PSP [7]. Nonetheless, CCG and C&P members were not included in the study and, therefore, data on mental health disorder prevalence and related sex differences among CCG and C&P remains sparse.

There is limited mental health research including CCG and C&P officers. The most relevant information relies heavily on research conducted with the United States Coast Guard (USCG). Previous research has reported 15% of USCG participants met the diagnostic criteria for PTSD, and 5% met the diagnostic criteria for MDD [11], whereas 19.0% met the criteria for a co-morbid diagnosis of PTSD and MDD. In a previous study of international seafarers, 25% of seafarers reported symptoms consistent with MDD, and 17% reported symptoms consistent with GAD [12]. In 2018, the United States (US) Department of Defense reported symptoms of psychological distress in the past year (10.6%) and past month (6.0%) were observed among members of the USCG [13]. In total, 7.3% also screened positive for PTSD in the past year. The USCG members also reported an average of 0.56 days were missed and productivity was impaired on an average of 1.56 days due to mental health disorder symptoms. More recently, since the COVID-19 pandemic (starting March 2020), many USCG members have reported symptoms of MDD (20.7%), GAD (22.7%), PTSD (18.4%), and elevated stress (38.2%) [14]. The differences between male and female coast guard members were not reported in the previous studies.

Despite the recognition that the CCG and C&P include PSP, previous Canadian PSP research on duty-related mental health issues has yet to include these sectors. There is currently no published research regarding the prevalence of current mental health disorder symptoms among CCG and C&P, and no information on how CCG and C&P PSP compare to the general population and other Canadian PSP (i.e., municipal/provincial police, firefighters, paramedics, Royal Canadian Mounted Police, correctional workers, and dispatchers). The current study was designed to provide estimates of several mental health disorder symptoms that can: 1) provide the prevalence of positive screenings for mental health disorders among CCG and C&P PSP; 2) facilitate explicit comparisons between CCG and C&P PSP, the general population and other Canadian PSP; and 3) facilitate an examination of the differences across sociodemographic categories, including sex differences.

The results are intended to support the recommended Canadian National Action Plan, which includes ongoing increasingly robust research regarding post-traumatic stress injuries (PTSIs) in Canadian PSP [15] and contribute to the wider Canadian PSP literature. In line with previous research [7,16,17], the participating CCG and C&P PSP were expected to have mental health disorder symptoms and a prevalence of positive screenings for mental health disorders higher than in the general population. Previous research indicates that female PSP may report more mental health disorder symptoms than their male counterparts [7], based on several mediating environmental variables [18,19]; accordingly, we hypothesized that the females would report more mental health disorder symptoms than the males in the current study. Based on previous research, the participants with previous PSP or military experience [20] were expected to report more mental health disorder symptoms than those with no previous work experience, as a function of their previous vocational requirements and more frequent exposures to PPTEs [4,21].

## 2. Materials and Methods

### 2.1. Procedure

Data were collected using a web-based self-report survey available in English or French. The study was approved by the University of Regina Institutional Research Ethics Board (REB# 2021-003). The survey was based on a set of validated measures used in a previous study of PSP [4,6,7,16], and by the Public Health Agency of Canada [15,22], but collaboratively redesigned by the research team and the CCG and DFO team to ensure relevant variables were included. The survey was promoted and distributed by the CCG and DFO to member unions via emails, social media posts, and a video encouraging participation. The survey was available from 1 February 2021 to 31 January 2022. At the start of the survey, the participants selected to complete the survey in English or French and were presented with the study information and an informed consent form. After consenting to participate, the participants were provided with a randomly generated unique code, which allowed for repeated survey access to complete the survey over multiple sessions. Participation was anonymous and no contact or identifying information was collected. To further protect anonymity, sample sizes of 4 or less were not reported. The current study focused specifically on the self-reported symptoms of mental health disorders, and the data was cross-sectionally analyzed.

### 2.2. Data and Sample

The participants were CCG/DFO members (*n* = 412) (67.5% CCG members and 26.0% C&P members). The responses from 561 CCG/DFO members were initially collected. Only the data from the respondents who completed at least 30% of the survey were retained. The final sample included in the current study analyses and results was a total of 412 members. The participants were mainly male (56.1%), identifying as men (55.3%), white (i.e., Caucasian) (82.8%), aged 30 to 39 years old (26.9%), or 40 to 49 years old (26.5%) (see Table 1). The participants were mostly married or in common-law relationships (i.e., living with a person in a conjugal relationship for 12 continuous months) (63.8%), with a college (37.4%) or a university (31.3%) degree, residing in British Columbia (53.2%), with no previous experience as either PSP or Canadian Armed Forces (CAF) (67.5%).

### 2.3. Self-Report Mental Health Disorder Symptom Measures

The survey asked participants to self-report symptoms related to various mental disorders. A ‘positive screen’ on any of the following measures indicated that the individual self-reported symptoms were consistent with expectations for a diagnosis of a particular mental health disorder. A positive screen on a self-report survey is not necessarily synonymous with meeting diagnostic criteria, which requires a clinical interview by a licensed professional. However, substantial differences in rates when comparing self-reported mental disorder symptoms consistent with a positive screen and interview assessments were not identified in a recent meta-analysis [23]. Nonetheless, the individuals who used the self-report measures and indicated a positive screen would require the evaluation of a trained and licensed clinician for the possible diagnosis of a specific mental health disorder. The current study assessed symptoms related to screening positive for the mental disorders of PTSD, MDD, GAD, SAD, Panic Disorder (PD), and Alcohol Use Disorder (AUD).

The PTSD Checklist for DSM-5 (PCL-5) [24,25] assessed for symptoms related to PTSD. PTSD involves intense or prolonged psychological distress at exposure to internal or external cues that symbolize or resemble an aspect of the traumatic event(s) [26]. For the PCL-5, participants rated how bothered they had been by 20 common symptoms of PTSD in the past month on a five-point Likert scale, from 0 (i.e., not at all) to 4 (i.e., extremely). The participants reported their behaviours over the past month. For the PCL-5, a positive screen required the participants to report exposure to at least one item from the Life Events Checklist for the DSM-5 (LEC-5) [27], meet the minimum DSM-5 criteria [26] for each PTSD symptom cluster subscale (e.g., intrusions, avoidance, negative alterations in cognitions and mood, and alterations in arousal and reactivity), and exceed the clinical cut-off of >32 [25]. A psychometric evaluation of the PCL-5 has demonstrated strong internal consistency (α = 0.94) and good test–retest reliability (r = 0.82) within populations exposed to PPTEs [24].

The 9-item Patient Health Questionnaire (PHQ-9) [28] assessed symptoms of MDD. A positive screen for MDD outlined by the DSM-5 requires the participants to report experiencing five or more symptoms during the same two-week period, with one of the symptoms being a depressed mood or a loss of interest or pleasure [26]. The symptoms must also cause the individual significant distress or impairment in social, occupational, or other areas of functioning, and cannot be due to substance use or another medical condition [26]. For the PHQ-9, the participants indicated how bothered they had been by depressive symptoms in the past two weeks by responding to each item using a four-point Likert scale (i.e., 0 = not at all to 3 = nearly every day). For the PHQ-9, a positive screen is indicated by a score of >9 [7]. A psychometric evaluation of the PHQ-9 has demonstrated good internal consistency (α = 0.89) and test–retest reliability (r = 0.84) within the general population [28].

The Panic Disorder Symptoms Severity Scale, Self-Report (PDSS-SR) [29] assessed symptoms of PD. PD involves experiencing an abrupt surge of intense fear or discomfort that reaches a peak within minutes and during which time four or more symptoms occur (i.e., palpitations, sweating, trembling, shortness of breath, feeling of choking, chest pain or discomfort, nausea, dizziness, etc.) [26]. For the PDSS-SR, the participants first read the definition of a panic attack and the accompanying symptoms. From the accompanying symptoms, at least four had to be endorsed (e.g., rapid or pounding heartbeat, sweating, nausea, feeling of choking) for a panic attack to have occurred. If the participant reported having ever experienced a panic attack or experienced a panic attack in the past week, they were asked additional questions rated on a five-point Likert scale (i.e., 0 = none to 4 = extreme). A positive screen required the PDSS-SR total scores to be >7 [29]. The self-report version has demonstrated excellent psychometrics with a strong internal consistency (α = 0.92) and an intraclass correlation coefficient of 0.81 [30].

The 7-item Generalized Anxiety Disorder Scale (GAD-7) [31] assessed symptoms of GAD. GAD involves (1) excessive anxiety and worry, occurring more days than not for at least six months, about a number of events or activities; (2) the individual finds it difficult to control the worry; (3) the anxiety and worry are associated with three or more symptoms (i.e., restlessness, fatigue, difficulty concentrating, irritability, muscle tension, sleep disturbance); (4) the anxiety and worry cause clinically significant distress or impairment in social, occupational, or other areas of functioning; and (5) the disturbance is not related to substance use, another medical condition, or better explained by another medical disorder [26]. For the GAD-7, the participants indicated the extent to which seven symptoms of anxiety bothered them in the previous two weeks. Ratings were made on a four-point Likert Scale (i.e., 0 = not at all to 3 = nearly every day). A positive screen for GAD required the GAD-7 total score to be >9 [31]. The GAD-7 has shown good reliability and construct, criterion, procedural, and factorial validity [31], as well as good internal consistency (α = 0.89) and inter-item correlations of 0.45–0.65 in a community sample [32].

The 14-item Social Interaction Phobia Scale (SIPS) assessed for symptoms of SAD [33]. According to the DSM-5, SAD involves (1) marked fear or anxiety about one or more social situations in which the individual is exposed to possible scrutiny by others; (2) social situations almost always provoke fear or anxiety; (3) the fear or anxiety is out of proportion to the actual threat posed by the social situation; (4) social situations are avoided or endured with intense fear or anxiety; (5) the fear, anxiety, or avoidance is persistent, typically lasting for six months or more; and (6) the fear, anxiety, or avoidance is not attributable to substance use, another medical condition, or better explained by another medical condition [26]. The SIPS includes three subscales to assess social interaction anxiety, fear of overt evaluation, and fear of attracting attention, respectively. Each item is rated on a five-point Likert Scale (i.e., 0 = not at all characteristic of me to 4 = entirely characteristic of me). There is no specific time window used. A positive screen for SAD requires a SIPS total score to be >20 [33]. The SIPS has demonstrated overall excellent internal consistency (α = 0.92), and convergent and discriminant validity in a large and independent sample [34].

The ten-item Alcohol Use Disorders Identification Test (AUDIT) assessed for symptoms of AUD [35]. According to the DSM-5, AUD involves an individual experiencing at least two symptoms related to drinking behaviors and negative consequences (i.e., drinking more or longer than you intended, trying to cut down or stop drinking but can’t, spending a lot of time drinking or hungover, distracted by wanting to drink so badly, drinking or being sick from drinking interfering with responsibilities, etc.) [26]. For the AUDIT, the participants were asked questions about their drinking behaviors and negative alcohol-related consequences. The ratings were made using Likert scales that varied across the items. A positive screen for AUD required the total AUDIT score to be >15 [28,31,36]. A psychometric evaluation of the AUDIT has demonstrated good internal consistency (α = 0.81), good test–retest reliability (r = 0.83 to 0.95) within the general population, and (α = 0.81) in a police-specific population [37,38,39].

### 2.4. Statistical Analyses

The participants were grouped into sociodemographic categories for comparisons of the positive screens for mental health disorders. The descriptive analyses, including frequencies and percentages of the sociodemographic variables (i.e., sex, gender, age, marital status, ethnicity, province of residence, education, previous work experience and occupation category), and the means and standard deviations for mental health symptom screening measures, are presented in Table 1. The prevalence of positive screens based on the self-reported mental health disorder measures was calculated as percentages. A series of t-tests and one-way analysis of variance (ANOVA) tests were performed to assess for statistically significant differences across sociodemographic categories and participant groups (e.g., total, CCG and C&P samples) based on the mean scores for the mental health disorder symptoms measures. All the tests were two-tailed with an alpha level of 0.05. Holm–Bonferroni adjustments were applied to the alpha levels in the post hoc tests to reduce the familywise error rate. Binomial tests were conducted to compare the prevalence of positive screens of mental health disorders between the current total, CCG, and C&P samples, and the general population [40,41,42,43,44,45]. A series of logistic regression models was conducted to test the association between sex and positive screens of mental health disorders among the total sample, as well as the CCG and C&P samples. All the data were analyzed using SPSS v.28 Premium (IBM, 2021 New York, NY, USA).

## 3. Results

The differences across the sociodemographic categories on the mental health disorder symptom measures are presented in Table 1. The females reported statistically significantly higher scores for symptoms of GAD (*p* < 0.05), SAD (*p* < 0.001), and PD (*p* < 0.01). The participants who reported previous work experience in the CAF and as PSP reported statistically significantly higher symptom measure scores for PTSD (*p* < 0.01), than those who had no previous work experience. No statistically significant differences in the mental health disorder symptom measure scores were observed between the CCG and C&P samples (see Table 1). Statistically significant effects were observed for age (*p* < 0.05), marital status (*p* < 0.05), education (*p* < 0.05), and ethnicity (*p* < 0.001) on SAD; age, and marital status (*ps* < 0.05) on PTSD; and ethnicity (*p* < 0.01) on PD. However, follow-up multiple pairwise comparisons were not statistically significant, due to the application of Holm–Bonferroni adjustments to the alpha levels in the post hoc tests to control familywise error rates.

The associations between sex and positive screens for mental health disorders are presented in Table 2. Based on the self-report measures, the female participants were more likely than the males to screen positive for SAD in the total sample (OR, 2.08; 95% CI, 1.25–3.45), and CCG sample (OR, 1.91; 95% CI, 1.06–3.47). The females were also more likely to screen positive for GAD in the C&P category (OR, 3.33; 95% CI, 1.05–10.55).

The prevalence percentages of positive screens for current mental health disorders for the total, CCG, and C&P samples are presented in Table 3. Where applicable, the prevalence percentages for the general population [40,41,42,43,44,45] are also presented for comparison. The prevalence of positive screenings of PTSD, MDD, GAD, SAD, PD, and AUD in the total sample was statistically significantly higher than in the general population (*ps* < 0.001). The participants (42.0%) self-reported a statistically significantly higher prevalence of symptoms related to positive screening for one or more current mental health disorders than in the general population (10.1%; *p* < 0.001) [40]. Comparisons between the current sample and previously published Canadian PSP were also conducted. The participants reported a statistically significantly lower prevalence of PTSD and a higher prevalence of SAD than the previously published Canadian PSP (*ps* < 0.001) [7].

## 4. Discussion

The current study helps fill important extant gaps in the literature on PSP mental health and presents the first known empirical evidence on a large, national, diverse sample of CCG and C&P PSP mental health; specifically, the prevalence of mental health disorder symptoms and positive screens for mental health disorders assessed using broadly accepted and validated screening measures. Understanding CCG and C&P officers’ mental health issues may provide important insights that can inform efforts to protect their mental health, reduce the impact of PPTEs and occupational stressors, and extend their years of service. As expected, among CCG and C&P officers, the prevalence of positive screenings of PTSD, MDD, GAD, SAD, PD, and AUD was statistically significantly higher than in the general population [40,41,42,43,44,45]. The prevalence of positive screens for a mental health disorder was also expected to be similar for CCG and C&P officers and other Canadian PSP; however, some differences were observed. CCG and C&P officers reported a statistically significantly lower prevalence of PTSD and a statistically significantly higher prevalence of SAD than previously published Canadian PSP [7].

The prevalence of positive screens for one or more mental health disorders among CCG and C&P PSP (42.0%) is much higher than the prevalence of diagnosed mental health disorders in the general population (10.1%) [40]. The current results indicate that CCG and C&P PSP are facing mental health challenges more than the general public, implying that the observed mental health challenges among CCG and C&P PSP may likely result from their cumulative service experiences which includes more frequent exposure to PPTEs [4] and occupational stressors [46] during their work. The prevalence of positive screens for one or more mental health disorders among CCG and C&P PSP was also comparable to the prevalence among other Canadian PSP (44.5%) [7]. The current results evidence similar mental health challenges among the groups and highlight the importance of including CCG and C&P officers in PSP PTSI research. The current results also provide evidence that CCG and C&P leaders and supervisors need to understand the mental health challenges of their personnel. The current results could be used to inform interventions to address the specific mental health disorders that may be impacting CCG and C&P members the most.

CCG and C&P PSP reported a statistically significantly lower prevalence of PTSD and a statistically significantly higher prevalence of SAD than previously published Canadian PSP [7]. The prevalence of positive screens on MDD, GAD, PD, and AUD among CCG and C&P were comparable to other Canadian PSP. In the current sample, CCG and C&P PSP had the highest positive screens for MDD (24.5%) and SAD (21.4%). The results differ from previously published Canadian PSP, who reported the highest positive screens for MDD (26.4%) and PTSD (23.2%) [7]. Differences in the prevalence of mental health disorders were also observed between the CCG and C&P groups. CCG PSP reported a higher percentage (17.1%) of positive screens for PTSD than the C&P sample (11.2%). However, no statistically significant differences were observed for mental health disorder symptom measure scores between the CCG and C&P groups. Differences in prevalence for specific mental health disorders between CCG, C&P, and other Canadian PSP suggest variables specific to each occupation differentially affect members.

The prevalence differences for specific mental health disorders between CCG and C&P samples and between CCG and C&P PSP and other Canadian PSP may be due to diverse factors, including differences in occupational experiences and activities, and exposures to PPTEs [4] and occupational stressors [46]. For example, CCG and C&P officers deploy frequently and to remote locations [47], which have less access to professional support [48,49]. There are also fewer CCG and C&P personnel spread across Canada, increasing the likelihood that CCG and C&P officers may work independently and therefore have less access to social supports [49,50], which may increase their odds of SAD [50]. The PPTE-type frequencies also differ; whereas PSP have previously reported most frequent exposures to sudden violent death (93.8%) and sudden accidental death (93.7%) [4], CCG and C&P officers reported most frequent exposures to serious transportation accidents (77.4%), and serious accidents at work, home, or during recreational activity (69.7%) [5]. CCG and C&P also differ in their occupational duties. C&P members may engage in duties specific to law enforcement and the protection of species at risk, whereas CCG engage in duties specific to marine assistance and national security [3,22,47]. The prevalence and range of positive screens of mental health disorders among CCG and C&P underscore the need to identify the diverse risk and resiliency factors that may help to address the observed mental health challenges.

There were several sociodemographic factors associated with positive screens for any mental health disorder. Female participants reported statistically significantly higher scores for symptoms of GAD and PD. Female participants were also more likely than males to screen positive for SAD. This was observed in the CCG sample but not in the C&P sample. In the C&P sample, females were more likely to screen positive for GAD. Sex differences were expected, and the current results align with evidence that female PSP are more likely than males to report mental health disorders [7]. GAD, SAD, and PD are anxiety-based mental health disorders. Researchers have shown that anxiety differs among male and female police officers depending on their occupational duties [51]. Female PSP may be exposed to more sexualization, disrespect, sexually charged threats, and violence than their male counterparts [52]. Work-related anxiety among female CCG and C&P members could be an explanation for the higher prevalence of anxiety disorders observed in the current study.

Differences in behavioral mental disorder risk factors (i.e., sleep, social support, and stress) may also provide an explanation for the current results. The results are consistent with a study of Canadian police officers that observed sex was indirectly related to GAD, SAD, PD, PTSD, and MDD through its relationship with social support and sleep quality [53]. CCG and C&P PSP must manage operational (e.g., risk of injury on the job, paperwork, stigma from the public) and organizational stressors (e.g., staff shortages, lack of resources, favoritism). Compared to their male counterparts, females report being more negatively influenced by operational stressors [53]. Pressures may also exist for female PSP to act stoically on the job, which may compromise their coping or their overall resilience [52]. Accordingly, male and female CCG and C&P officers may have very different experiences of work-related stressors, despite having the same occupation. Increased levels of social support and sleep quality have been reported to be associated with decreased mental disorder symptoms [19]. Social support including both personal (e.g., family, friends) and organizational support (e.g., colleagues, commanding officers) and sleep may differ and impact male and female CCG and C&P differently, producing distinct implications for their mental health. However, less is known about these variables among CCG and C&P. Future research should examine how diverse variables differentially affect female CCG and C&P members and other PSP groups [7,8,9,10], and these variables should be highlighted when implementing mental health solutions.

The participants who reported previous work experience in both the CAF and working as PSP reported statistically significantly higher symptom measure scores for PTSD than those who had no previous work experience. The current results are consistent with what was expected. The positive screens for potential mental health disorders were reported to increase as a function of participant age and years of experience [7], as well as due to previous vocational requirements [22] in the CAF, or as PSP, and more frequent exposures to PPTEs [4,21]. Additionally, previous research has reported that, among other Canadian PSP, those with previous CAF experience reported significantly more exposures to PPTEs and were approximately 1.5 times more likely to screen positive for all mental disorders and report suicide ideation [54]. The combination of previous CAF and ongoing PSP service seems to increase the likelihood of screening positive for mental health disorders. Identifying and understanding the diverse impact of CAF experience on the mental health of members currently serving as PSP is necessary to help these individuals succeed in their careers and remain healthy after their military service.

Overall, the current study provides the first known national information on mental health disorder symptoms and positive screenings among CCG and C&P members. The selected measures allow for comparisons with other large occupational studies designed to estimate mental health disorders symptoms in specific occupational samples, such as PSP [7,16,17], and contribute to the Canadian PSP data. The current results provide potentially important information to support researchers investigating possible ways to mitigate and manage PTSI among PSP as part of the Canadian National Action Plan. The current results may also inform wider occupational health research initiatives, such as the U.S. National Institute for Occupational Safety and Health’s Total Worker Health Framework. Specifically, the current results evidence the impact of the work environment on the mental health of workers and may inform interventions and training developed and implemented to mitigate and manage mental health challenges among workers.

### Strengths and Limitations

The use of data provided by a national and diverse sample of CCG and C&P personnel is an important strength of the current study; however, several limitations should be noted and provide direction for future research. First, the survey was promoted and distributed by the CCG and DFO to personnel via emails, internal newsletters, internal webpages, videos, and posters encouraging participation. The CCG and C&P include approximately 6700 members (i.e., CCG 6100, C&P 600). The current sample reflects approximately 6.15% of CCG and C&P members and includes larger proportions of CCG members (67.5%) than C&P members (26.0%). The current sample also includes relatively larger proportions of members from British Columbia, and smaller proportions of members from Quebec, Nova Scotia, Newfoundland, and Labrador. However, the participant sociodemographic information indicates the sample was generally proportionally consistent with the age and sex of the CCG and C&P personnel. Nevertheless, the current sample may not be entirely representative of the entire CCG and C&P workforce.

Second, participation in the current study was anonymous, voluntary, and self-selected. The recruitment materials described the study as focusing on mental health disorders, PPTEs, occupational stress, and burnout, which may have attracted participants who were experiencing clinically significant mental health symptoms. The recruitment materials may partially explain the differences in prevalence rates between the current sample and the general population; however, the CCG and C&P members who were experiencing the most severe symptoms may have been on leave, missed the invitation, or been too symptomatic to respond to a lengthy survey. Stigmatizing attitudes about mental health may also have inhibited some individuals from accessing the survey, despite assurances of anonymity. Additionally, the collection method using an online survey may have impacted the number of participants. Many CCG and C&P members do not have easy access to computers or the internet as they serve on ships, stations, or in the field, and are often away for long periods of time. The participants were able to begin, leave, and return to the survey at their leisure, to ease the survey response burden; as such, there is no way to know the average length of survey completion time or to understand why some participants (6.5%) did not complete the entire survey. The data were also collected over 12 months from 1 February 2021 to 31 January 2022. The extended data collection period and extenuating circumstances of the COVID-19 pandemic may have impacted the results of the current study.

Third, the screening measures for mental health disorders used in the current study are valid and reliable for use in clinical settings; nevertheless, diagnoses can only be made using clinical interviews with supporting collateral information. The participants reported their current symptoms as assessed by the screening measures, with time periods ranging from 7 days to the past year. Further, only a relatively small number of potential mental health disorders were screened for in the current study. The frequency of positive mental health disorder screens lends support to the need for additional research using Statistics Canada sampling methods and clinical interviews to make more reliable assessments and to allow for comparisons with the general population. Clinical interviews assessing lifetime prevalence would also help to discern whether symptoms developed prior to or over the course of the participants’ careers. The inclusion of self-reported past mental health disorder diagnoses prior to and since starting a PSP career would provide some insight into the participants’ lifetime mental health.

## 5. Conclusions

The current results offer the first known empirical evidence of CCG and C&P PSP mental health, specifically the prevalence of mental health disorder symptoms and positive screens for mental health disorders among a national and diverse sample of CCG and C&P officers. The results indicate that many of the CCG and C&P PSP screened positive for clinically significant symptom clusters consistent with one or more mental health disorders. The prevalence of positive screens among the CCG and C&P was much higher than the diagnostic rates for the general population, highlighting that mental health challenges are present among CCG and C&P PSP and may likely result from their service experiences. There were also significant differences between the CCG and C&P and other Canadian PSP, underscoring the need for further investigation into the context around diverse risk and resiliency factors that may help to address the observed mental health challenges. Overall, the current results provide insightful information into the mental health challenges facing CCG and C&P PSP and inform efforts to mitigate and manage PTSI among PSP.

## Figures and Tables

**Table 1 ijerph-19-15696-t001:** Categorical Participant Sociodemographics and Mean Scores on Mental Health Disorder Screening Measures.

	Total Sample ^2^	PTSD		MDD		GAD		SAD		PD		AUD	
	*% (N)*	*Mean (*SD*)*	*n*	*Mean (*SD*)*	*n*	*Mean (*SD*)*	*n*	*Mean (*SD*)*	*n*	*Mean (*SD*)*	*n*	*Mean (*SD*)*	*n*
Total Sample													
	100(412)	18.64(16.47)	369	6.47(5.78)	383	4.98(4.59)	369	11.17(11.24)	364	1.94(3.97)	357	6.57(5.69)	329
Gender													
Man	55.3(228)	16.88(16.39)	218	6.15(5.93)	226	4.47(4.45)	217	9.35(10.11) ^b^	217	1.43(3.51) ^b^	213	7.12(5.41)	194
Woman	36.7(151)	20.30(16.13)	143	6.77(5.49)	149	5.64(4.63)	144	13.47(11.98) ^a^	139	2.60(4.49) ^a^	136	5.94(6.10)	127
Non-Binary	1.2(5)	35.80(11.82)	5	9.20(3.27)	5	6.60(3.85)	5	19.60(9.29) ^a,b^	5	4.80(3.42) ^a,b^	5	4.60(1.94)	5
Two-Spirits	-	-		-	-	-	-	-	-	-	-	-	-
Test Statistic ^1^	-	F(3365) = 4.74 **		F(3379) = 1.56		F(3365) = 2.57		F(3360) = 5.81 ***		F(3353) = 3.66 *		F(3325) = 2.55	
$\mathrm{Effect Size}\;({𝜂}_{p}^{2})$	-	0.038		0.012		0.021		0.046		0.030		0.023	
Sex													
Male	56.1(231)	17.12(16.54)	221	6.16(5.90)	229	4.46(4.42)	220	9.47(10.16)	220	1.45(3.51)	216	7.02(5.38)	197
Female	37.4(154)	20.65(16.07)	146	6.87(5.48)	152	5.69(4.62)	147	13.54(11.93)	142	2.66(4.47)	139	5.97(6.09)	130
Test Statistic ^1^	-	t(365) = −2.02		t(379) = −1.18		t(365) = −2.57 *		t(360) = −3.47 ***		t(353) = −2.84 **		t(325) = 1.64	
Effect Size (Cohen’s d)	-	0.216		0.123		0.273		0.374		0.308		0.185	
Age													
19–29	11.9(49)	14.52(14.29)	48	5.22(5.11)	49	4.61(4.08)	49	10.73(10.96)	49	1.43(2.58)	49	6.56(4.80)	45
30–39	26.9(111)	16.98(14.34)	105	6.84(5.94)	110	5.27(4.32)	105	11.86(11.40)	106	1.35(2.98)	99	7.56(6.48)	93
40–49	26.5(109)	22.65(17.46)	106	7.04(5.49)	108	5.81(4.95)	102	13.21(12.13)	99	2.62(4.52)	97	6.69(5.34)	90
50–59	22.3(92)	18.60(18.03)	85	6.43(6.16)	91	4.34(4.64)	90	9.74(10.91)	88	2.07(4.46)	89	5.98(5.83)	82
60 +	5.1(21)	17.00(17.80)	20	5.10(6.48)	20	3.28(4.78)	18	5.24(3.78)	17	2.44(5.86)	18	3.50(2.78)	16
Test Statistic ^1^	-	F(4359) = 2.67 *		F(4373) = 1.21		F(4359) = 2.08		F(4354) = 2.49 *		F(4347) = 1.54		F(4321) = 2.12	
$\mathrm{Effect Size}\;({𝜂}_{p}^{2})$	-	0.029		0.013		0.023		0.027		0.017		0.026	
Education													
High School or Less	8.5(35)	18.74(17.14)	35	4.57(5.73)	35	3.43(4.37)	32	7.06(9.35)	31	1.66(3.97)	32	6.36(6.09)	28
College Program (e.g., Trade School; 2-Year College Diploma)	37.4(154)	19.02(16.79)	148	6.77(5.87)	152	5.26(4.65)	146	10.85(10.28)	143	1.92(3.85)	139	6.95(6.07)	123
Coast Guard College: Graduated Fleet	9.5(39)	14.05(12.63)	38	5.77(5.17)	39	4.23(4.09)	39	8.31(7.28)	39	0.68(1.89)	38	6.05(5.01)	37
Coast Guard College: MCTS Officer Training	2.2(9)	15.13(17.54)	8	4.67(2.96)	9	4.22(2.86)	9	15.67(14.34)	9	1.22(2.44)	9	8.29(6.73)	7
University Degree (4-year College or Higher)	31.3(129)	18.55(16.53)	121	6.73(5.78)	128	5.10(4.36)	125	12.34(12.46)	125	2.31(4.31)	121	6.74(5.57)	117
Test Statistic ^1^	-	F(4345) = 0.80		F(4358) = 1.50		F(4346) = 1.57		F(4342) = 2.50*		F(4342) = 1.39		F(4307) = 0.33	
$\mathrm{Effect Size}\;({𝜂}_{p}^{2})$	-	0.009		0.016		0.018		0.028		0.016		0.008	
Ethnicity													
Asian	2.2(9)	18.44(18.53)	9	8.00(7.98)	9	7.11(7.30)	9	13.75(15.74)	8	5.13(5.46)	8	2.86(3.24)	7
Black	^	^	^	^	^	^	^	^	^	^	^	^	^
Hispanic	^	^	^	^	^	^	^	^	^	^	^	^	^
Indigenous (i.e., First Nations, Inuit, Métis)	3.2(13)	16.69(15.06)	13	5.92(6.13)	13	4.31(4.85)	13	11.69(13.28)	13	1.46(3.86)	13	5.60(4.35)	10
South Asian	^	^	^	^	^	^	^	^	^	^	^	^	^
White	82.8(341)	17.99(15.92)	323	6.23(5.51)	337	4.77(4.26)	324	10.39(10.17)	321	1.74(3.63)	315	6.83(5.86)	291
Prefer Not to Answer	1.2(5)	30.20(20.05)	5	11.20(9.58)	5	9.20(8.96)	5	^	^	7.00(10.95)	5	5.20(4.44)	5
Other	3.6(15)	30.53(23.84)	15	10.20(7.78)	15	7.93(6.63)	14	17.21(17.44)	14	3.92(5.17)	13	4.17(3.19)	12
Test Statistic ^1^	-	F(7361) = 1.61		F(7375) = 1.65		F(73,361) = 2.01		F(7356) = 4.18 ***		F(7349) = 2.69 **		F(7321) = 0.97	
$\mathrm{Effect Size}\;({𝜂}_{p}^{2})$	-	0.030		0.030		0.038		0.076		0.051		0.021	
Marital Status													
Single	20.6(85)	17.66(16.35)	83	6.20(6.31)	84	4.99(4.56)	79	13.47(13.25)	79	1.96(3.70)	79	7.28(6.83)	72
Married/Common Law	63.8(263)	17.90(16.33)	251	6.28(5.59)	261	4.86(4.62)	253	10.53(10.42)	251	1.79(4.00)	242	6.40(5.28)	226
Separated/Divorced	7.3(30)	25.77(16.17)	26	8.24(5.23)	29	5.21(3.97)	28	8.12(8.35)	25	2.33(3.28)	27	7.05(5.96)	22
Widowed	^	^	^	^	^	^	^	^	^	^	^	^	^
Test Statistic ^1^	-	F(3358) = 2.82 *		F(3372) = 1.12		F(3358) = 0.06		F(3353) = 3.06 *		F(3346) = 0.18		F(3318) = 0.82	
$\mathrm{Effect Size}\;({𝜂}_{p}^{2})$	-	0.023		0.009		0		0.025		0.002		0.008	
Province of Residence ^3^													
British Columbia	53.2(219)	19.00(16.38)	210	6.42(5.93)	215	5.00(4.77)	204	11.02(11.25)	205	1.89(3.99)	197	6.19(5.69)	191
New Brunswick	1.7(7)	20.71(11.91)	7	6.00(5.60)	7	5.43(3.64)	7	8.83(9.28)	6	1.71(2.36)	7	6.20(2.68)	5
Newfoundland and Labrador	6.8(28)	18.89(15.17)	27	6.04(5.27)	28	5.61(4.53)	28	10.57(11.68)	28	1.96(3.57)	28	6.75(4.86)	24
Nova Scotia	9.0(37)	19.22(19.60)	36	6.08(5.68)	37	4.86(4.74)	37	9.72(8.77)	36	2.62(5.04)	37	6.79(6.77)	28
Ontario	10.9(45)	20.24(16.29)	42	7.73(6.01)	45	5.27(4.18)	44	14.90(11.94)	42	2.37(4.37)	41	7.88(6.17)	41
Québec	11.9(49)	14.62(15.94)	45	6.20(5.16)	49	4.11(4.28)	47	9.96(11.98)	45	1.18(2.82)	45	6.56(4.94)	39
Test Statistic ^1^	-	F(6361) = 0.74		F(6375) = 0.44		F(6361) = 0.61		F(6356) = 1.08		F(6349) = 0.58		F(5322) = 0.62	
$\mathrm{Effect Size}\;({𝜂}_{p}^{2})$	-	0.012		0.007		0.010		0.018		0.010		0.010	
Previous Work Experience													
Neither	67.5(278)	17.50(15.78)^b^	263	6.32(5.67)	274	5.06(4.50)	263	11.51(11.18)	259	1.85(3.68)	254	6.76(6.05)	232
Public Safety Only	15.8(65)	18.56(17.01) ^a,b^	64	6.31(5.76)	65	4.73(4.75)	64	10.38(11.75)	64	1.79(3.97)	62	6.28(4.64)	61
CAF Only	8.3(34)	23.00(18.14) ^a,b^	32	7.68(6.38)	34	5.09(5.17)	33	9.91(11.41)	32	3.06(6.06)	32	6.11(4.96)	28
CAF and Public Safety	2.4(10)	35.20(16.66) ^a^	10	7.70(7.07)	10	4.11(4.28)	9	11.33(9.67)	9	1.67(2.18)	9	4.75(4.62)	8
Test Statistic ^1^	-	F(3365) = 4.67 **		F(3379) = 0.72		F(3365) = 0.20		F(3360) = 0.32		F(3353) = 0.94		F(3325) = 0.47	
$\mathrm{Effect Size}\;({𝜂}_{p}^{2})$	-	0.037		0.006		0.002		0.003		0.008		0.004	
Occupation Category													
CCG	67.5(278)	18.98(16.94)	262	6.52(5.83)	275	5.20(4.71)	262	11.84(11.38)	257	2.10(4.17)	251	6.55(5.65)	231
C&P	26.0(107)	17.54(15.31)	105	6.26(5.69)	106	4.36(4.23)	105	9.57(10.82)	105	1.54(3.44)	104	6.66(5.82)	96
Test Statistic ^1^	-	t(365) = 0.76		t(379) = 0.39		t(365) = 1.58		t(360) = 1.75		t(232) = 1.32		t(325) = −0.15	
Effect Size (Cohen’s d)	-	0.087		0.044		0.183		0.202		0.142		0.018	

Notes. -: n = 0; ^: Sample size between 1 and 4, so data not presented. AUD = Alcohol Use Disorder; CAF = Canadian Armed Forces; CCG = Canadian Coast Guard; C&P = Conservation and Protection; GAD = Generalized Anxiety Disorder; MCTS = Marine Communications and Traffic Services; MDD= Major Depressive Disorder; PD = Panic Disorder; PTSD = Post-Traumatic stress disorder; SAD= Social Anxiety Disorder. ^1^ The test results comparing scores on mental disorder screening measures across categorical participant demographics. ^2^ Total percentages may not sum to 100 and *ns* may not sum to 412 due to non-response or responding “other.” ^3^ Province of residence: No values to report for Alberta and the Northern Territories (Yukon, Northwest Territories, Nunavut); ^a,b^ lettered superscripts within each column category indicate significant differences between category groups with different letters on outcome at *p* ≤ 0.05. * *p* < *0*.05, ** *p* < *0*.01, *** *p* < *0*.001–Statistically significantly different. Holm–Bonferroni adjustment applied to alpha levels to control Type I errors. Post hoc tests were not performed for some of the significant tests because at least one group had fewer than two cases.

**Table 2 ijerph-19-15696-t002:** Associations Between Sex and Mental Health Disorders.

		PTSD	MDD	GAD	SAD	PDD	AUD
		OR (95% CI)	OR (95% CI)	OR (95% CI)	OR (95% CI)	OR (95% CI)	OR (95% CI)
Total Sample	Male	1	1	1	1	1	1
	Female	1.16(0.66, 2.02)	1.33(0.83, 2.14)	1.43(0.81, 2.52)	2.08(1.25, 3.45) **	1.74(0.83, 3.65)	0.94(0.41, 2.15)
CCG	Male	1	1	1	1	1	1
	Female	1.19(0.63, 2.24)	1.37(0.78, 2.41)	1.11(0.57, 2.16)	1.91(1.06, 3.47) *	1.69(0.75, 3.83)	0.71(0.27, 1.86)
C&P	Male	1	1	1	1	1	1
	Female	0.66(0.17, 2.59)	1.18(0.48, 2.92)	3.33(1.05, 10.55) *	2.54(0.94, 6.88)	1.46(0.23, 9.20)	2.10(0.40, 11.07)

Notes. OR = Odds Ratios; CI = Confidence Interval. AUD = Alcohol Use Disorder; CAF = Canadian Armed Forces; CCG = Canadian Coast Guard; C&P = Conservation and Protection; GAD = Generalized Anxiety Disorder; MDD = Major Depressive Disorder; PD = Panic Disorder; PTSD = post-traumatic stress disorder; SAD = Social Anxiety Disorder; * *p* < 0.05, ** *p* < 0.01, statistically significantly different.

**Table 3 ijerph-19-15696-t003:** Current Mental Disorder Prevalence Rates Based on Self-Report Measures.

Disorder	General Population	Total Sample	Comparing Prevalence among Total Sample and General Population	CCG	Comparing Prevalence among CCG and General Population	C&P	Comparing Prevalence among C&P and General Population
	%	% (*n*)	Test Statistic	% (*n*)	Test Statistic	% (*n*)	Test Statistic
AUD	3.2 ^1^	7.9(26)	4.69 ***	8.7(20)	4.53 ***	6.3(6)	1.41
GAD	5.9 ^2^	15.7(58)	7.89 ***	16.4(43)	7.09 ***	13.3(14)	3.03 ***
MDD	2.2 ^3^	24.5(94)	29.64 ***	23.6(65)	24.03 ***	26.4(28)	16.67 ***
PD	1.5 ^4^	9.0(32)	11.38 ***	10.8(27)	11.81 ***	4.8(5)	2.37 **
PTSD	1.7 ^1^	14.7(60)	20.10 ***	17.1(47)	19.51 ***	11.2(12)	7.24 ***
SAD	3.2 ^6^	21.4(78)	19.61 ***	22.6(58)	17.47 ***	19.0(20)	8.95 ***
Any Anxiety Disorder	4.7 ^5^	30.1(113)	23.11 ***	31.5(84)	20.52 ***	26.2(28)	10.26 ***
Any Mood Disorder	5.4 ^1^	24.5(94)	16.46 ***	23.6(65)	13.25 ***	26.4(28)	9.36 ***
Any Disorder	10.1 ^1^	42.0(173)	21.40 ***	45.3(126)	19.39 ***	42.1(45)	10.81 ***
Total Number of Positive Screens							
0	-	58.0(239)	-	54.7(152)	-	57.9(62)	-
1	-	19.9(82)	-	21.6(60)	-	19.6(21)	-
2	-	10.7(44)	-	10.8(30)	-	12.1(13)	-
3 or More	-	11.4(47)	-	12.9(36)	-	10.3(11)	-
Total Sample	100	100(412)					

Notes. AUD = Alcohol Use Disorder; CCG = Canadian Coast Guard; C&P = Conservation and Protection; GAD = Generalized Anxiety Disorder; MDD = Major Depressive Disorder; PD = Panic Disorder; PTSD = post-traumatic stress disorder; SAD = Social Anxiety Disorder; ^-^ No data available. ^ Sample size between 1 and 4, so data not presented. ** *p* < 0.01, *** *p* < 0.001. References: ^1^ [40] ^2^ [41] ^3^ [42] ^4^ [43] ^5^ [44] ^6^ [45].

## Data Availability

Not applicable.

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
