# Peer review of "Mental Health Disorder Symptoms among Canadian Coast Guard and Conservation and Protection Officers"

_ijerph, 2022, doi:10.3390/ijerph192315696_

Round 1

Reviewer 1 Report

I have two comments that I would like the authors of the manuscript to react to:

1) The survey was available between February 1, 2021, to January 31, 2022. It was a time when many changes in the market and in the personal lives have taken place. I believe that the results of the study may have been affected by the specific time, context of COVID-19 and circumstances at the time. For example, people may have experienced more anxiety and bad mood in spring 2021 than winter 2022. Therefore, it is incorrect to analyze the survey results obtained in spring 2021 and winter 2022 as homogeneous. How would the authors justify that such a wide period of data collection does not influence the responses of the study participants. How were side variables controlled for in the study? Did the authors of the study compare the data of questionnaires completed in different periods?

2) I would suggest the authors not only to inform about the obtained results, but also to provide an interpretation of the obtained results. I especially missed it in the Discussion chapter.

Author Response

1.1 I have two comments that I would like the authors of the manuscript to react to:

1.2. The survey was available between February 1, 2021, to January 31, 2022. It was a time when many changes in the market and in the personal lives have taken place. I believe that the results of the study may have been affected by the specific time, context of COVID-19 and circumstances at the time. For example, people may have experienced more anxiety and bad mood in spring 2021 than winter 2022. Therefore, it is incorrect to analyze the survey results obtained in spring 2021 and winter 2022 as homogeneous. How would the authors justify that such a wide period of data collection does not influence the responses of the study participants. How were side variables controlled for in the study? Did the authors of the study compare the data of questionnaires completed in different periods?

Thank you for your feedback to strengthen the manuscript. We appreciate your time and effort spent to review the manuscript and hope you will find our updates satisfactory and the manuscript suitable for publication.

Thank you for your comments regarding the data collection period. The impact of time is always a potential challenge. An argument can also be made that a data collection that occurs on a single day does not account for fluctuations that occur over time. In any given year there are always many environmental changes that can occur such that collecting over a year may actually provide a more robust representation. There is certainly an argument to be made that comparing pre-COVID data/results to post-COVID data/results may be inappropriate, but COVID was well underway during 2021. Nevertheless, based on your concerns analyses were conducted to examine for differences between date of survey completion and scores on the mental health disorders symptoms measures. There were only two statistically significant relationships between day completed and PHQ-9 (r(381) = .102, p <.05.) and GAD-7 (r(367) = .138, p <.01) scores. The results could indicate that people who participated in the study later reported statistically significantly higher PHQ-9 and GAD-7 scores, which may be important; however, we have no way of knowing whether the relationships are spurious. To address your concerns and the ad hoc analyses, we have added the data collection period to the limitation as a possible influence of the results. However, it is possible that this pattern exists in the general population and does not differ from the norm.

Additionally, regarding confounding variables. Confounding variables were controlled for using GLM-Univariate effect sizes with the ANOVAs.

1.3. I would suggest the authors not only to inform about the obtained results, but also to provide an interpretation of the obtained results. I especially missed it in the Discussion chapter.

Thank you for your comment and opportunity to strengthen the manuscript. We have updated the discussion to further interpret the obtained results and provided implications and recommendations based on the results.

Reviewer 2 Report

The introduction is too large, too detailed, has too many points, and needs summarised introduction 

Demographics - Whites, common law relationship > what does it mean?

Why did you use t-Tests, and why not others, why ANOVA 

How did you control confounders 

How many people participated

why some did not participate 

Did people exaggerate their problems to get benefits 

Why did you choose the population to study 

Why do female has more problem

whats the gender-based job satisfaction rate

Does this work needs muscle strength, does the work needs more physical power 

how did you protect confidentiality of the worker 

whats the consenting process 

discussion needs rewriting with some headings 

what's the experience of gender-based discrimination in the work 

any cultural discrimination causing the issue 

why did you use those scales? needs to explain each of that scale 

Is there any previous study in this population or any qualitative research 

Author Response

Thank you for your feedback to strengthen the manuscript. We appreciate your time and effort spent reviewing the manuscript and hope you will find out update satisfactory and the manuscript suitable for publication.

2.1. The introduction is too large, too detailed, has too many points, and needs summarised introduction

Thank you for your suggestion. Updates have been made to the introduction to present the information as succinctly as possible without compromising the flow and presentation of the information.

2.2. Demographics - Whites, common law relationship > what does it mean?

Thank you for identifying this opportunity to clarify. We have updated the sentence to include (i.e.,) explanation for white (i.e., Caucasian) and common-law relationships (i.e., living with a person in a conjugal relationship for 12 continuous months).

2.3. Why did you use t-Tests, and why not others, why ANOVA 

T-tests and ANOVAs were selected to compare categories means. This method is consistent with previous PSP research and with Statistics Canada.  

2.4. How did you control confounders 

Thank you for your comment. Confounding variables were controlled for using GLM-Univariate effect sizes with the ANOVAs.

2.5. How many people participated

Explanation about participation in the study is available in the Data and Sample section. Responses from 561 CCG and C&P members were initially collected. Only data from respondents who completed 30% of the survey were retained. The final sample included in the current study analyses and results was a total of 412 CCG and C&P members.

2.6. why some did not participate 

Information about why some CCG and C&P members did not participate is mentioned in the limitations. Due to anonymous participation and the use of access codes which allowed the participants to complete the survey at their leisure, makes it difficult to understand factors that may have affected participation (i.e., time to complete the survey)

2.7. Did people exaggerate their problems to get benefits 

Though not impossible, this is unlikely. PSP already have excellent benefits packages that go unused, and PSP are more likely to underreport symptoms of mental disorders due to greater levels of mental health stigma and organizational stigma towards those with mental disorders. It is also likely that members who were experiencing severe mental health disorder symptoms were on leave and did not participate. We have addressed both of these possibilities in the limitations.

2.8. Why did you choose the population to study 

The Canadian Coast Guard and the Department of Fisheries and Oceans include members (CCG members and C&P Officers) with duty-specific responsibilities similar to other professions that have been categorized as PSP. Despite the recognition of CCG and C&P officers as PSP, they have yet to be included in PSP specific research on mental health disorders. We have selected to examine mental health disorders among CCG and C&P Officers to fill the gap in the literature, to contribute to the current data on Canadian PSP, and to support the Canadian National Action Plan that includes ongoing robust research regarding posttraumatic stress injuries in Canadian PSP.

2.9. Why do female has more problem

Thank you for your question. We have taken the opportunity to update in the discussion section about this result. The paragraph regarding sex differences (pg. 13 and 14) has been expanded to include additional explanations and for the differences between males and females.

2.10. whats the gender-based job satisfaction rate

Thank you for your question. Unfortunately, we have no way to answer this question with the currently available data.

2.11. Does this work needs muscle strength, does the work needs more physical power 

Thank you for your question. The amount of physical strength required would vary in any given day, but all employees would be required to meet a minimum set of employment requirements.

2.12. how did you protect confidentiality of the worker 

Thank you for identifying this opportunity to clarify on confidentially. Additional information about anonymity of the survey has been added to the procedures along with a statement about how we protected anonymity when presenting the data.

2.13. whats the consenting process

Thank you for identifying this opportunity to clarify about the consenting process. Additional information about obtaining consent has been added to the procedures section.  

2.14. discussion needs rewriting with some headings

Thank you for your suggestion. To strengthen the discussion, we have made updates to throughout the section but did not think it was necessary to add sub-headings

2.15. what's the experience of gender-based discrimination in the work 

Thank you for your question. Unfortunately, we have no way to answer this question with the currently available data.

2.16. any cultural discrimination causing the issue 

We examined sociodemographic differences and no statistically significant differences were observed for ethnicity. The sample was mostly white (i.e., Caucasian), so this could account for lack of significant differences. We have addressed this in the limitations that the sample may not be representative of the entire CCG and C&P workforce.

2.17. why did you use those scales? needs to explain each of that scale

Thank you for identifying an area to strengthen the manuscript. Information about the set of validated measures used in the survey is available in the procedure section. These measures were selected to be consistent with previous studies of Canadian PSP and the Public Health Agency of Canada. Information about each validated mental health disorder symptoms measure has been added to the self-reported measures section, including reliability and consistency data.

2.18. Is there any previous study in this population or any qualitative research 

To the best of our knowledge this is the first study including Canadian Coast Guard and Conservation and Protection Officers.

Reviewer 3 Report

This is a review of the manuscript titled "Mental Health Disorder Symptoms Among Canadian Coast Guard and Conservation and Protection Officers". The purpose of this study is to examine the prevalence of mental health disorders among Canadian Coast Guard (CCG) and Conservation and Protection (C&P) Officers. It was found that the positive screens for mental health disorders of CCG and C&P Officers were more prevalent than the general population. Females were found to be more likely than males to have social anxiety disorder and generalized anxiety disorder. This manuscript could be improved if the following concerns are addressed:

1. The authors mention a number of mental disorders such as posttraumatic stress disorder (PTSD), major depressive disorder (MDD, panic disorder (PD), generalized anxiety disorder (GAD), social anxiety disorder (SAD), and alcohol use disorder (AUD). I suggest the authors provide definitions or brief descriptions of these mental disorders.

2. It is stated that, "In a diverse sample of Canadian PSP, 44.5% screened positive for any mental health disorder 23.2% screened positive for PTSD, and 26.4% screened positive for MDD." (p. 2) However, it is later stated that, "previous Canadian PSP research on duty-related mental health issues has yet to include these sectors." (p. 2). It is confusing. Did the previous study using a diverse sample of Canadian PSP included CG and C&P Officers?

3. It is stated that, "Previous research indicates that female PSP may report more mental health disorder symptoms than their male counterparts" (p. 3) As the investigation of gender differences in mental disorder symptoms is an important research objective of this study, the authors should provide explanation for those gender differences.

4. Please describe the scoring format for each self-report instruments.

Author Response

Thank you for your feedback to strengthen the manuscript. We appreciate your time and effort spent reviewing the manuscript and hope you will find out update satisfactory and the manuscript suitable for publication.

3.1. The authors mention a number of mental disorders such as posttraumatic stress disorder (PTSD), major depressive disorder (MDD, panic disorder (PD), generalized anxiety disorder (GAD), social anxiety disorder (SAD), and alcohol use disorder (AUD). I suggest the authors provide definitions or brief descriptions of these mental disorders.

Thank you for your comment and opportunity to strengthen the manuscript. Information has been added about the DSM-5 definition or criteria for each of the disorders mentioned in the current study. Information has also been added about each of the mental health disorder symptom measures including information on scoring, consistency and reliability.

3.2. It is stated that, "In a diverse sample of Canadian PSP, 44.5% screened positive for any mental health disorder 23.2% screened positive for PTSD, and 26.4% screened positive for MDD." (p. 2) However, it is later stated that, "previous Canadian PSP research on duty-related mental health issues has yet to include these sectors." (p. 2). It is confusing. Did the previous study using a diverse sample of Canadian PSP included CG and C&P Officers?

Thank you for identifying this opportunity to clarify. We have updated the paragraph to include who the diverse sample of Canadian PSP are and clarify that CCG and C&P were not included in the previous study.

3.3. It is stated that, "Previous research indicates that female PSP may report more mental health disorder symptoms than their male counterparts" (p. 3) As the investigation of gender differences in mental disorder symptoms is an important research objective of this study, the authors should provide explanation for those gender differences.

Thank you for identifying this opportunity to strengthen the manuscript. Information about sex differences has been added to the introduction. The paragraph discussing the current results related to sex differences (pg. 13 and 14) has also been expanded to include possible explanations for those differences.

3.4. Please describe the scoring format for each self-report instruments.

Thank you for this suggestion. Information about each validated mental health disorder symptoms measures has been added to the self-reported measures section, including scoring, reliability, and consistency data.

Reviewer 4 Report

Hello dear authors!

Your research is interesting and informative.

I have a few questions that I would like an answer to

1. Has your questionnaire been standardized?

2. Why were only men included in the study, and how does the absence of women in the sample affect the sensitivity of testing?

A few notes on the bibliography

1. It is necessary to complete the link in full (there is no imprint) - number 11, 27, 33,

Author Response

Your research is interesting and informative.

Thank you for your positive comment and feedback to strengthen the manuscript. We appreciate your time and effort spent reviewing the manuscript and hope you will find our updates satisfactory and the manuscript suitable for publication.

I have a few questions that I would like an answer to

4.1. Has your questionnaire been standardized?

Thank you for your question and opportunity to clarify the manuscript. The survey included a set of validated measures (PCL-5, PHQ-9, PDSS-SR, GAD-7, SIPS, and AUDIT) which were also used in previous PSP studies and are consistent with the measures used by the Public Health Agency of Canada. This statement appears in the procedures section. Information about each of the measures and their scoring, reliability, and consistency has been added to the self-report measures section.

4.2. Why were only men included in the study, and how does the absence of women in the sample affect the sensitivity of testing?

We apologize for the confusion, both male and females were included in the study. We have updated the abstract to include the percentage of females to clarify that both are included in the current study. The current study also examined sex differences of mental health disorder prevalence among Canadian Coast Guard and Conservation and protection officers.

4.3. A few notes on the bibliography

1. It is necessary to complete the link in full (there is no imprint) - number 11, 27, 33,

Thank you for identifying these errors. We have corrected the references.

Reviewer 5 Report

Please see attached my document with my comments.

Author Response

This manuscript describes an important gap in the occupational stress literature and focused on an understudied working population such as those working in the Coast Guard and Protection Officers.

Thank you for your positive comments and feedback to strengthen the manuscript. We appreciate your time and effort spent reviewing the manuscript and hope you will find our updates satisfactory and the manuscript suitable for publication.

5.1. Abstract: Overall, the abstract is easy to follow. However, the authors should highlight a bit more about the implications of their research for future practice and wellness interventions for this population of workers.

Thank you for your suggestion. We have updated the abstract to include a statement about implications of the current results for interventions and training for CCG and C&P.

5.2. Page 2, paragraph 3 (Introduction): for line 65, there is a comma missing “4.5% screened positive for any mental health disorder 23.2% screened” should be “4.5% screened positive for any mental health disorder, 23.2% screened.”

Thank you for identifying this mistake. The sentence has been corrected.

5.3. Page 2, paragraph 5 (Introduction): This statement seems out of place and should be included in the discussion section: “The current study overcomes such limitations with a large, national, diverse sample of CCG and C&P PSP assessed using broadly accepted and validated screening measures. Understanding CCG and C&P officers mental health issues may provide important insights that can inform efforts to protect their mental health, reduce the impact of PPTEs and occupational stressors and extend their years of service.”

Thank you for this suggestion. We agree the statement would be better suited for the discussion and have moved it.

5.4. General comment about introduction: The introduction needs work. There is no theoretical framework that is mentioned that may have guided this research study. I recommend that the authors ground their research on theory. There are many theoretical perspectives that can be adopted in order to better understand the mental health stressors in this population of workers. In addition, the introduction needs to be structures for better flow. It is unclear what the hypotheses are that the authors are testing. They are listed in the introduction, but should be described more clearly in a logical fashion supported by previous research and a theoretical framework.

Thank you for your comments and opportunity to strengthen the manuscript. Updates have been made to the introduction to succinctly summarize some information, to improve the flow, and better connect the current research to the hypothesis. The current work is meant to be an empirical, rather than theoretical, piece of research; as such, there would be broad latent theoretical bases that could be defined, but that would go well beyond the scope of the current work. Pending editorial request, we can add a such a theoretical paragraph to the introduction.

5.5. Page 3, paragraph 1 (Methods): The authors should clarify in the methods section whether this was a cross-sectional data-set/study.

Thank you for the suggestion. We have added this information to the methods section.

5.6. Page 3-4, paragraph 4 (Methods): The authors should specify the Cronbach’s alpha for the questionnaires used since they mention that they used validated instruments to capture the data.

Thank you for identifying this opportunity to strengthen the manuscript. Information about each of the validate mental health disorder symptom measures has been added to the self-report measures section including information about scoring, reliability and consistency.

5.7. Page 2, paragraph 1 (Discussion): The authors should also discuss their findings within the context of a theoretical framework. This was also missing in the introductory section.

The current work is meant to be an empirical, rather than theoretical, piece of research; as such, there would be broad latent theoretical bases that could be defined and exposed, but that would go well beyond the scope of the current work. Pending editorial request, we can add a such a theoretical paragraph to the introduction and the discussion, but we feel that would be inconsistent with the intent of the work.

5.8. Finally, I recommend that the authors include a section in the conclusion section about the implications of their findings and its relevance to the U.S. National Institute for Occupational Safety and Health’s Total Worker Health Framework – specifically looking at work as a social determinant of health and the conditions of the work environment that impacts the mental health of this population of workers. I understand that this is a population of workers residing in Canada, but I do think that a discussion of the Total Worker Health framework in relation to the study findings would be impactful for readers.

Thank you for this suggestion and opportunity to connect the current research and results to wider occupational research initiatives. We have added a paragraph at the end of the discussion that mentions how the current results may inform the U.S. National Institute for Occupational Safety and Health’s Total Worker Health Framework.

5.9. I was also looking for the authors to make future recommendation and the potential implications of their findings in terms of wellness interventions for this population of workers. I did not see that discussion included. Otherwise, the discussion section was very well written. Great work!

Thank for identifying this area to strengthen the manuscript and presentation of the current results. We have added additional recommendations about the implication of the current results.

Round 2

Reviewer 2 Report

GOOD WORK 

Reviewer 3 Report

The manuscript has been improved and is acceptable for publication in the current form.